# Effects of COVID-19 lockdown on heart rate variability

**Nicolas Bourdillon**[1,2]*, **Sasan Yazdani**[2], **Laurent Schmitt**[3], **Grégoire P. Millet**[1]

**1** Institute of Sport Sciences, University of Lausanne, Lausanne, Switzerland, **2** be.care SA, Renens, Switzerland, **3** National Centre of Nordic-Ski, Research and Performance, Prémanon, France

* nicolas.bourdillon@unil.ch

## Abstract

### Introduction

Strict lockdown rules were imposed to the French population from 17 March to 11 May 2020, which may result in limited possibilities of physical activity, modified psychological and health states. This report is focused on HRV parameters kinetics before, during and after this lockdown period.

### Methods

95 participants were included in this study (27 women, 68 men, 37 ± 11 years, 176 ± 8 cm, 71 ± 12 kg), who underwent regular orthostatic tests (a 5-minute supine followed by a 5-minute standing recording of heart rate (HR)) on a regular basis before (BSL), during (CFN) and after (RCV) the lockdown. HR, power in low- and high-frequency bands (LF, HF, respectively) and root mean square of the successive differences (RMSSD) were computed for each orthostatic test, and for each position. Subjective well-being was assessed on a 0–10 visual analogic scale (VAS). The participants were split in two groups, those who reported an improved well-being (WB+, increase >2 in VAS score) and those who did not (WB-) during CFN.

### Results

Out of the 95 participants, 19 were classified WB+ and 76 WB-. There was an increase in HR and a decrease in RMSSD when measured supine in CFN and RCV, compared to BSL in WB-, whilst opposite results were found in WB+ (i.e. decrease in HR and increase in RMSSD in CFN and RCV; increase in LF and HF in RCV). When pooling data of the three phases, there were significant correlations between VAS and HR, RMSSD, HF, respectively, in the supine position; the higher the VAS score (i.e., subjective well-being), the higher the RMSSD and HF and the lower the HR. In standing position, HRV parameters were not modified during CFN but RMSSD was correlated to VAS.

### Conclusion

Our results suggest that the strict COVID-19 lockdown likely had opposite effects on French population as 20% of participants improved parasympathetic activation (RMSSD, HF) and

**Data Availability Statement:** Data underlying the study is available on Zenodo (http://doi.org/10.5281/zenodo.4058116).

**Funding:** NB and SY are employees of becare SA. GPM and LS have no conflict of interests. becare SA did not play a role in the study design, data

collection and analysis, decision to publish, or preparation of the manuscript and only provided financial support in the form of NB and SY salaries. The funder provided support in the form of salaries for authors NB and SY, but did not have any additional role in the study design, data collection and analysis, decision to publish, or preparation of the manuscript. The specific roles of these authors are articulated in the 'author contributions' section.

**Competing interests:** NB and SY are employees of becare SA. GPM and LS have no conflict of interests. becare SA did not play a role in the study design, data collection and analysis, decision to publish, or preparation of the manuscript and only provided financial support in the form of NB and SY salaries. The funder provided support in the form of salaries for authors NB and SY, but did not have any additional role in the study design, data collection and analysis, decision to publish, or preparation of the manuscript. The specific roles of these authors are articulated in the 'author contributions' section. NB and SY were employed by the company be.care SA. The remaining authors declare that the research was conducted in the absence of any commercial or financial relationships that could be construed as a potential conflict of interest. This does not alter our adherence to PLOS ONE policies on sharing data and materials.

rated positively this period, whilst 80% showed altered responses and deteriorated well-being. The changes in HRV parameters during and after the lockdown period were in line with subjective well-being responses. The observed recordings may reflect a large variety of responses (anxiety, anticipatory stress, change on physical activity. . .) beyond the scope of the present study. However, these results confirmed the usefulness of HRV as a non-invasive means for monitoring well-being and health in this population.

## Introduction

The COVID-19 pandemic in 2020 resulted in strict confinement rules in France for 8 weeks, from 17 March to 11 May, period known as the lockdown. Its aim was to slow down the spread rate of COVID-19, therefore saving lives and decreasing the workload in hospitals, which capacity to treat new patients was exceeded. Restrictions during the lockdown included closed gyms and sports centers, plus limitations on the duration spent outdoors, maximum walking distance from home which, associated with the lack of space and appropriate devices in homes for physical exercise, and lack of technical knowledge of the population on appropriate training routines, may have resulted in decreased physical activity. A survey by the French public health national institute "Santé Publique France" [1] published on 17[th] June 2020 confirmed that 47.4% of the population declared a decrease in physical activity, especially (58.9%) in walking time, during the lockdown period. Contradictory, a minority (17.9%) used this period for increasing their physical activity. In addition, 33.4% of the population declared spending more than 7h seated. The average daily seated time was 6h19. Beyond these self-declared data, the impact of the lockdown period remains unclear, strong psychological impacts have been reported in a review [2] and specifically to the COVID-19-related lockdown [3], as well as metabolic and body composition outcomes [4], but one may hypothesize that it may have led to changes in cardiovascular fitness and psychological states.

Exercise plays a fundamental role in cardiovascular health [5] but also in mitigating some of the possible psychological impacts of lockdown [6]. In those cases, aerobic exercises such as walking or endurance running appears to be more beneficial than sprint running or force/power exercises [6, 7]. Heart rate variability (HRV) is a commonly used method for cardiovascular follow-up in athletes [8], healthy people [9] cardiovascular patients [10], and is related to self-regulations at the cognitive, emotional, social and health levels [11, 12]. Specifically, HRV analysis allows to evaluate the modulation of the sympathetic and parasympathetic branches of the autonomic nervous system. The low-frequency (LF) band reflects a mix of sympathetic and parasympathetic modulations on the heart [13] and has been proposed as an index of the sympathovagal balance [14]. The migration of the respiratory sinus arrhythmia in the low-frequency band is a subject of debate [15], and it is not the only issue regarding the physiological meaning of LF [16], which leaves the physiological implications of this frequency band unclear [17]. Respiratory sinus arrhythmia is the main phenomenon provoking changes in the HF band and RMSSD which therefore mainly reflects parasympathetic influences on the heart [18, 19]. Physical exercise increases the parasympathetic tone [20]. Hypotonia is a sign of cardiovascular deconditioning [21]. Finally, HRV-based methods may be used to assess autonomic imbalances, diseases [10], stress and well-being [12].

To the best of our knowledge, HRV parameters before, during and after the lockdown period have not been reported in the literature. Therefore, the aim of the present study was to report HRV parameters to demonstrate the effects of lockdown and to investigate whether changes in HRV were related to some changes in psychological states during and after this

period. We hypothesized that alteration in parasympathetic-related markers (i.e., decreases in RMSSD and HF, increase resting HR) would reflect a negative feeling over the period, whilst a minority of the population would report an improved subjective feeling with consequences on HRV, potentially in line with the report by Santé Publique France [1].

## Methods

### Experimental design

Strict lockdown (further denoted CFN) rules were applied in France over 8 weeks, from 17 March to 11 May 2020. The lockdown was unexpected, therefore there was no previous intention to study its effects on HRV. Nevertheless, data collection was performed for the purpose of this study, using recording devices distributed to the participants before the lockdown. The equipment consisted of a RR-interval-measuring chest strap and a dedicated mobile application that recorded both RR intervals from the strap and a declared visual analogic scale (VAS). When accepting to use the mobile app, the participants accepted the participation in the study. Three periods were compared: From 1 January to 16 March (baseline, BSL); from 17 March to 11 May (CFN), and from 12 to 31 May (recovery, RCV). The local ethical committee approved the study (agreement 2016–00308; Commission Cantonale d'Ethique de la Recherche sur l'être humain, CCER-VD; Lausanne, Switzerland) as part of a set of studies on a broader scale. All experimental procedures conformed to the standards set by the Declaration of Helsinki.

### Participants

Inclusion criteria were, age between 18 and 60 years old. All subjects self-declared being physically active, healthy, non-smokers, with no known diseases and no pregnancy or lactation for women as well as living in France. The initially planned study involved physical exercises with a maximum age for inclusion criteria of 60 years-old to limit the cardiovascular risks. Out of the database (n = 112), participants with at least forty orthostatic tests between 1 January and 31 May 2020 and at least 10 tests in each period of three periods (BSL, CFN and RCV) were extracted (n = 95). More specifically, from January 1st to May 11th (end of lockdown) there were 20 weeks. With 2 tests per week in average per participant, 40 tests were performed.

### Heart rate variability

The participants performed an orthostatic test (5 minutes supine followed by 5 minutes in the standing position) on a regular basis between the 1st of January 2020 and the 31st of May 2020 [22]. This orthostatic test had to be performed, in the morning, upon wake-up, with an empty bladder, and before breakfast. The inter-beat interval (RR interval) measuring device (sensor H7 + chest belt, Polar, Kempele, Finland or cardiosport TP5+, Cardiosport, Waterlooville, UK) was connected to the participants' smartphones via Bluetooth (mobile application: inCORPUS® v2.1.3, be.care SA, Renens, Switzerland). The H7 is the most used RR-recording device, the TP5+ uses the exact same technology [23]. The smartphone application is a Bluetooth receiver, using the standard BLE protocol, which does not interfere in any ways with the recording.

The participants were walked through the test by the dedicated mobile application, they started the recording from the application, in the supine position. After five minutes the smartphone beeped, at which point the participants immediately stood and pressed a button on the application to start the recording of the standing phase. After five minutes in the standing phase the smartphone beeped again to inform that the test was over. At that point, the participants were asked to fill-up the VAS by sliding the cursor according to their sensation of the

moment. The transition phase between the supine and the standing positions last 3 to 8 s on average. The HRV analysis does not take into account the first minute of recording in each position. Data were immediately and automatically transferred to our servers for analysis. Therefore, meetings between participants and researchers were avoided, in the respect of the strict lockdown rules applying. Becare S.A. employees controlled that the participants properly used the mobile application and chest belt and responded quickly to any question from the participants using dedicated chat bots, emails or telephone.

Out of each of the 5-minute periods, the first minute was used as a stabilization period and was excluded from the analysis whilst the last four minutes were analyzed for HRV.

The RR intervals from the orthostatic tests were first analyzed to remove ectopic beats from the recordings, the algorithms used are currently not available to the public. Ectopic beats were then compensated for by means of interpolation to calculate normal-to-normal (NN) intervals. From the NN intervals, the following heart rate variability (HRV) parameters were extracted: mean HR; the root mean square of the successive differences (RMSSD); the spectral power in the low-frequency (LF, 0.04–0.15 Hz) and high-frequency bands (HF, 0.15–0.40 Hz) in $ms^2$; and the values (expressed in normalized units) for LF and HF, labelled nLF and nHF, respectively [8, 24]. The spectral power was estimated using a fast Fourier transform on the resampled NN intervals (4 Hz) using a window length of 250 data points and an overlap of 50%. All computations were performed separately for the supine (SU) and standing (ST) positions using. All analyses were performed using MATLAB® (R2019a, MathWorks, Natick, MA, USA).

### Visual analogic scale (VAS)

On the mobile application, for each test, the participants were required to rate their well-being "how do you feel today?" using a visual analogic scale (VAS) graduated from 0 to 10.

### Statistics

All results are given as mean ± SD. For display purposes SEM is plotted on all figures. HR and HRV units were normalized with respect to BSL. The Kolmogorov-Smirnov test was used to ensure normality of the data. Measurements were evaluated with a two-way ANOVA analysis for time effect (BSL, CFN and RCV) and group effect (WB- vs. WB+). Regarding the group effect division adopted (WB- vs WB+), the criteria was a change in VAS >2, that corresponded to 20% change in the scale. Such a large change aimed to lower the intra-individual variability. The statistical power of the performed tests was set at $p = 0.05$ for significance and $p = 0.10$ for tendency. The Tukey-Kramer post-hoc was used when appropriate. Statistical power SP is reported for each ANOVA performed, the computations take into account the F-statistic values (expected and observed) and the degrees of freedom [25, 26]. ANOVA analyses were carried out on unitary values (i.e. no average per period or otherwise). Correlations were computed for LF, HF, HR and RMSSD against VAS, in the supine and the standing positions, regardless of the time points (BSL, CFN and RCV), which means that all data points were pooled when computing the correlations. Correlations were made using MATLAB® (R2019a, MathWorks, Natick, MA, USA). Statistical analyses were performed using MATLAB®.

### Results

Ninety-five participants (27 women, 68 men) were included in this report (age 37 ± 11 years, height 176 ± 8 cm, weight 71 ± 12 kg). Out of these participants, 20% (WB+, n = 19; 5 women, 14 men; age 36 ± 13 years, height 178 ± 10 cm, weight 74 ± 13 kg) showed an improved well-being, estimated by the positive change (>2) in the 0–10 VAS score between BSL and CFN.

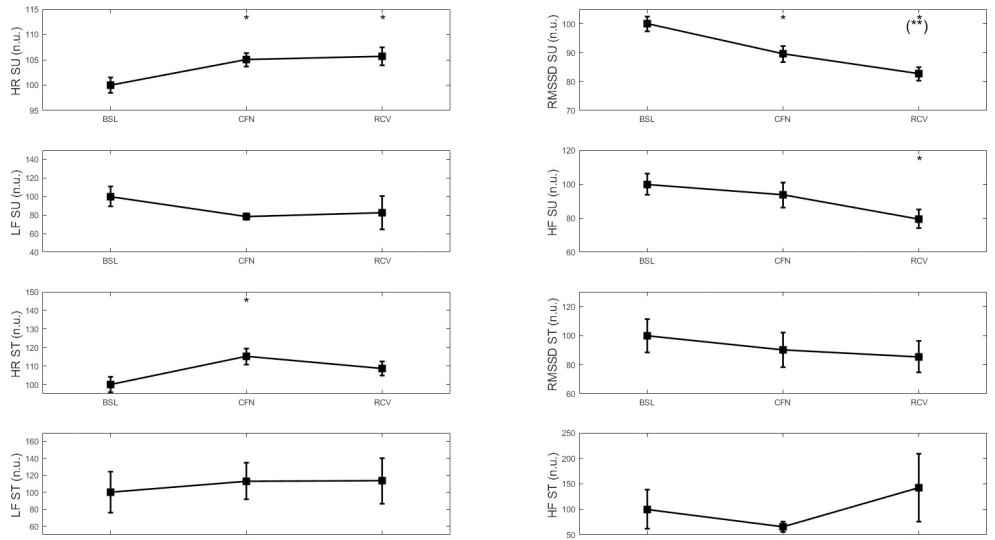

**Fig 1. HR, RMSSD, LF and HF before (BSL), during (CFN) and after (RCV) the strict lockdown period in France in 76 participants who deteriorated or maintained their well-being over the CFN period.** SU: supine position, ST: standing position. * $p < 0.05$, ** $p < 0.01$ for difference with BSL. # $p < 0.05$ for difference with CFN.

Contradictory, most of the participants (WB-, n = 76; 22 women, 54 men; age 38 ± 11 years, height 175 ± 7 cm, weight 69 ± 12 kg) did not improve or deteriorated this VAS score during the CFN period.

Fig 1 shows the negative impact of CFN in the WB- group, with a general alteration in parasympathetic activity in the supine position, as shown by the increased supine HR and decreased RMSSD in CFN and RCV compared to BSL (SP = 0.76). There was a tendency for a decrease in RMSSD between CFN and RCV (SP = 0.99). There was also a decrease in HF in RCV compared with BSL (SP = 0.48). In the standing position, there was a significant increase in HR in CFN compared to BSL (SP = 0.64), but there was no change on the other parameters. All these changes are reported in normalized units.

Fig 2 shows the positive impact of CFN in the WB+ group with a decrease in supine HR in CFN and RCV compared to BSL (SP = 0.85), an increase in supine RMSSD in CFN and RCV compared to BSL and an increase in RCV compared to CFN (SP = 0.99). There was also an increase in LF (SP = 0.36) and HF (SP = 0.98 and therefore total power) in RCV compared to BSL, and an increase in HF in RCV compared to CFN. There was no change in the standing position. All these changes are reported in normalized units.

There was an interaction time x group for HR, RMSSD and HF ($p < 0.001$, SP = 0.69; $p < 0.001$, SP = 0.90; and $p < 0.01$, SP = 0.65; respectively) in the supine position. There was a significant interaction for HR only ($p < 0.05$, SP = 0.42) in the standing position. In the supine position, HR was significantly higher in the WB- group compared to the WB+ group in CFN and RCV. RMSSD was significantly lower in the WB- group compared to the WB+ group in CFN and RCV. HF was significantly lower in the WB- group compared to the WB+ group in RCV. In the standing position HR was significantly higher in the WB- group compared to the WB+ group in CFN.

There were significant correlations ($p < 0.001$ for all) between VAS and HR ((R = -0.24), RMSSD (R = 0.14) and HF (R = 0.19) on the whole population in the supine position. The higher the VAS score (i.e., subjective well-being), the higher the RMSSD and HF and the lower

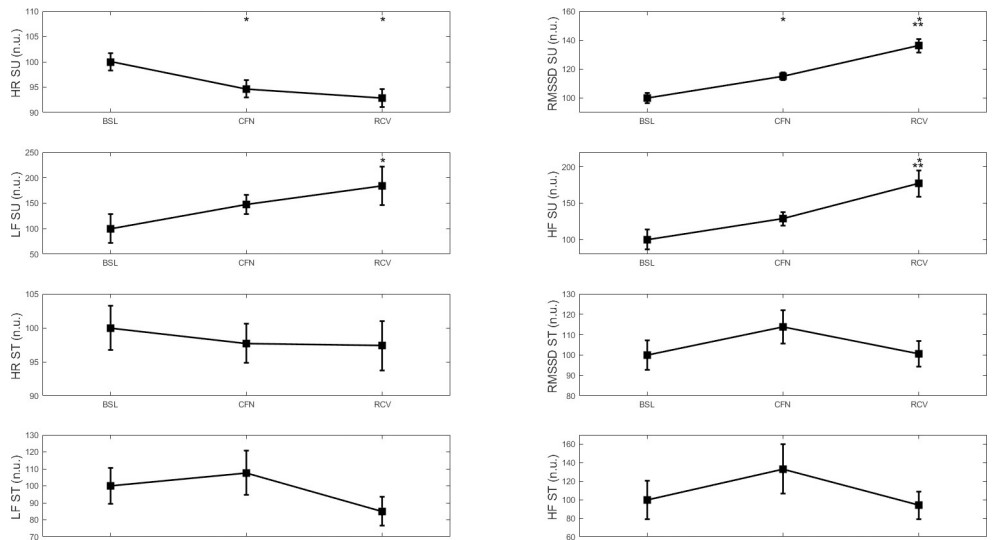

**Fig 2. HR, RMSSD, LF and HF before (BSL), during (CFN) and after (RCV) the strict lockdown period in France in 19 participants who improved their well-being over the CFN period.** SU: supine position, ST: standing position. * $p < 0.05$ ** $p < 0.01$ for difference with BSL. # $p < 0.05$ for difference with CFN.

the HR. Moreover, RMSSD in standing position was correlated to VAS. All Pearson's correlation data are listed in Table 1. All correlations were computed on absolute values.

When dividing the CFN period in two sub-periods of 4 weeks, there were no differences in any of the parameter between CFN1 and CFN2, suggesting a relative stability across CFN (S1 Data).

## Discussion

The present study aimed to investigate the HRV responses of participants exposed to the strict lockdown that occurred in France. We found that HRV changes from BSL to CFN and RCV were contradictory between two groups who did vs. did not deteriorate their subjective well-being over these periods. In 80% of the participants, parasympathetic-related HRV parameters (i.e., RMSSD, HF) measured in the supine position were decreased concomitantly to a psychological alteration. Contradictory, in 20% of the participants, both supine HRV and well-being were improved. Standing HRV was not modified during CFN.

The sudden lockdown that occurred implied drastic changes in the lifestyle of the participants, which is generally a stressor factor. Moreover, the psychological impact of lockdown included negative psychological effects, such as frustration, boredom, loss of social contacts, [2] inadequate supplies, inadequate information, financial loss, infection fears, confusion, and anger [27, 28]. The participants in the present study lived in France, where a survey of the French national public health institute (Santé Publique France) demonstrated a drastic decrease in physical activity and increased time spent in the seated position [1]. In the present study, physical activity was not measured, but only subjective feeling of well-being, measured by a VAS with a smartphone. This resulted in 20% of the participants who rated positively the lockdown period, since VAS score improved by at least 2 on the 0–10 scale. Interestingly, this is in line with data from the French national public health institute, who reported that 18% of the French population declared an increase in physical activity. Obviously, well-being can be determined by many causes and physical activity is only one of them. In this increasing well-being, the fact that increased RMSSD, LF and HF (associated to decreased HR) were also

**Table 1. Pearson's correlation coefficient (R), p value, and 95% confidence interval (CI) in the supine and standing positions for HR and HRV parameters against VAS.**

|  | R | p | CI |
|---|---|---|---|
|  | supine | | |
| LF (ms$^2$) | 0.07 | 0.08 | -0.01 0.15 |
| HF (ms$^2$) | 0.14 | < 0.001 | 0.06 0.22 |
| HR (bpm) | -0.24 | < 0.001 | -0.3–0.18 |
| RMSSD (ms) | 0.19 | < 0.001 | 0.11 0.27 |
|  | standing | | |
| LF (ms$^2$) | -0.01 | 0.89 | -0.12 0.10 |
| HF (ms$^2$) | 0.08 | 0.15 | -0.03 0.19 |
| HR (bpm) | -0.02 | 0.52 | -0.09 0.04 |
| RMSSD (ms) | 0.10 | 0.01 | 0.03 0.17 |

observed post lockdown, may be a marker of an effective and lasting improvement in cardio-vascular health. The best responses occur when both the sympathetic and parasympathetic influences on the heart are high [29, 30] hence increased LF and HF, associated to decreased resting HR.

Beyond this speculative influence of physical activity, there are many stressors during lockdown [2]: subjective (i.e., poor information on the duration of the lockdown, the fears of infection, frustration and boredom, sense of isolation, anger, guilt, helplessness, loneliness, nervousness, sadness) or objective (i.e., inadequate supplies, financial concerns, altered usual day-to-day activities etc.) sources of stress have an impact on psychological outcomes and indirectly on HRV.

There were significant correlations between VAS and HRV, which was positive for HF and RMSSD and negative for HR. Therefore, the better the participants felt, the higher the parasympathetic influences and the lower the HR, and vice-versa the lower the HF and RMSSD, the lower the feeling of well-being. Most of these correlations were found when HRV was measured in supine position, except for RMSSD in standing position. This later result emphasizes the value of RMSSD as a key parameter of HRV analysis. Moreover, these results emphasize the usefulness of HRV as a global index of heath in the general population. In line with previous literature and data from the national French public health institute, the more active people during the lockdown may have felt better [1, 2].

## Limitations

The lockdown was applied suddenly, and it was therefore impossible to recruit participants a priori during the baseline period. By chance we had enough participants with sufficient HRV tests to conduct this study. Overall, it is a unique opportunity to describe the HRV behavior in 95 healthy people, extracted from a database of 7098 orthostatic tests. The main limitation is that we did not have any data on physical activity of these participants. We can only speculate that the HRV changes during CFN reflect at least partly the decrease or increase in physical activity. Another important limitation of the present study is that our sample did not allow any other group divisions, evidently justifiable by literature, as by age clusters or gender that may be predictors of negative psychological impacts of lockdown. Moreover, Hnatkova et al. [31] showed that heart rate responses to postural changes may be different for sex. Finally, the VAS allowed to only rate a "general well-being" but not any specific mood components as vigor, anxiety, anger etc. Therefore, it remains vague and incorporates probably different psychological states among participants.

Through questionnaire in the application and for each test, the participants declared their potential lack of sleep, alcohol consumption (as limited as possible), digestive problems and other potential confounding factors that may have modified the HRV parameters reported in this study. Those factors are known as key in HRV and well-being alterations and may partly explain the cardiovascular deconditioning observed. However, in the present study, we did not find any factor that may have explained the differences between the two groups.

The separation criterion between the WB+ and WB- groups was arbitrary

## Conclusion

The lockdown in France was sudden and severe, inducing an increase in many subjective and objective stressors as well as drastically limiting the possibilities for physical activities. This likely resulted in a stressful situation that may lead to negative psychological outcomes and/or cardiovascular health (i.e., decreased RMSSD and HF and increased resting heart rate in supine position) in most of the population. However, in 20% of the participants the lockdown was positively rated, as shown by an improved well-being and parasympathetic activity. Altogether, the present study confirms the usefulness of regular HRV testing in this population for monitoring health status, as previously shown in elite athletes.

## Supporting information

**S1 Data.**
(TXT)

## Author Contributions

**Conceptualization:** Nicolas Bourdillon, Grégoire P. Millet.

**Data curation:** Sasan Yazdani.

**Formal analysis:** Nicolas Bourdillon, Sasan Yazdani.

**Investigation:** Nicolas Bourdillon.

**Methodology:** Nicolas Bourdillon, Sasan Yazdani.

**Supervision:** Laurent Schmitt, Grégoire P. Millet.

**Validation:** Grégoire P. Millet.

**Writing – original draft:** Nicolas Bourdillon.

**Writing – review & editing:** Nicolas Bourdillon, Sasan Yazdani, Laurent Schmitt, Grégoire P. Millet.

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
