## [Decision Letter · Decision Letter 0]

1 Sep 2020

PONE-D-20-22573

Effects of COVID-19 lockdown on heart rate variability

PLOS ONE

Dear Dr. Bourdillon,

Thank you for submitting your manuscript to PLOS ONE. After careful consideration, we feel that it has merit but does not fully meet PLOS ONE’s publication criteria as it currently stands. Therefore, we invite you to submit a revised version of the manuscript that addresses the points raised during the review process.

We look forward to receiving your revised manuscript.

Kind regards,

Daniel Boullosa

Academic Editor

PLOS ONE

Journal Requirements:

2. Please note that PLOS does not allow reference to data not shown (stated at the end of the results section). Thus, before we proceed, we kindly ask you provide the relevant data within the manuscript, the Supporting Information files, or in a public repository. If the data are not a core part of the research study being presented, please remove any references to these data.

3. During our internal evaluation we noted that participants provided verbal consent, did the ethics committees/IRBs approve this consent procedure? Please explain: i) Why was written consent not obtained?  ii) How did you record/document participant consent? Please provide these consent details in the ethics statement in the both the Methods section of the manuscript and in the online submission information.

4. Thank you for including your competing interests statement; "I have read the journal's policy and the authors of this manuscript have the following competing interests: Nicolas Bourdillon and Sasan Yazdani are employees of becare S.A."

5.  Please upload a copy of Figure 3, to which you refer in your text on page 5. If the figure is no longer to be included as part of the submission please remove all reference to it within the text.

Reviewers' comments:

Reviewer's Responses to Questions

**Comments to the Author**

1. Is the manuscript technically sound, and do the data support the conclusions?

Reviewer #1: Yes

Reviewer #2: Partly

2. Has the statistical analysis been performed appropriately and rigorously? 

Reviewer #1: Yes

Reviewer #2: No

3. Have the authors made all data underlying the findings in their manuscript fully available?

Reviewer #1: Yes

Reviewer #2: No

4. Is the manuscript presented in an intelligible fashion and written in standard English?

Reviewer #1: Yes

Reviewer #2: Yes

5. Review Comments to the Author

Reviewer #1: In this paper, the authors investigated changes in heart rate variability before, during and after the lockdown which took place for about two months in France.

Data was collected according to standard procedure (morning test), and all typical metrics are ported, in the time (rMSSD) and frequency (LF HF) domains, for a decent sample of 95 individuals

Most importantly, subjective data was collected together with objective physiological data, which I find paramount in this case considering that despite what much of the literature reports, physiology needs to be contextualised with other variables, as exemplified by the fact that HRV could go either way (assuming a similar response for all individuals would be naive), and indeed this change does relate with the objective measurement of physiological stress derived via HRV analysis

The sample gets a bit small when stratified (e.g. only 5 women in the WB+ group), however as the changes are analysed relatively to their own historical data (baseline), I would consider this issue less problematic than in the case of comparing groups

My speculation for the lack of change in the standing position might be linked to the protocol, often the orthostatic measurement creates more issues than it tries to resolve, with individuals getting anxious or too worked up by the quick change of posture, which might result in the test itself just capturing that transitory readjustment, which has little to do with baseline physiological stress - which is what the test aims at capturing. It could be helpful in this case simply to do two separate tests, maybe separated by a few minutes, but this procedure is of course less practical. Unfortunately in the paper it is not clear if the protocol required a short period between the two phases (e.g. less than 30 seconds) or not, and therefore if this speculation makes sense. Please revise the paper including more information about the exact protocol, and also discuss this aspect if applicable at that point.

Interesting to see also how the change in ANS activity was rather stable during CFN

I understand that this is not the focus of the paper, but it would be relevant for practitioners and users of these technologies if the authors could speculate on what is behind the differences, why a smaller group felt better (which was reflected in their ANS) and why a larger group had more issues, any how this second group could potentially make changes in case another lockdown happens, to favore a more positive response

Given that the data was collected as part of other studies, I wonder if any other potential correlates were collected (physical activity apparently not), but anything else that could shed some light on factors behind these differences would be relevant)

In this context I find the discussion section quite limited by the focus on physical activity. In my experience, large changes in heart rate and rMSSD baseline are hardly associated to changes in physical activity alone, especially over rather short time periods and in healthy individuals, while psychological stressors might have a much larger impact. Given that the lockdown is clearly a huge psychological stressor, I would expect more discussion points around this aspect.

Note also how your own data seems to highlight how the psychological aspect is key (how are you feeling is the question, which does not seem to be tuned to physical activity), more than the physical activity aspect. Of course lack of physical activity as per se a strong psychological effect especially in previously active people, but then the conversation is not about de-training, it's again about psychological wellbeing when unable to perform physical activity. Hopefully this makes sense to the authors and the discussion can be extended to better cover these aspects, which of course are all speculative since no data has been collected on other covariates

Reviewer #2: The manuscript presents recordings of heart rate variability (HRV) and subjective well-being in a sample of French healthy adults before, during and after a period of strict confinement, imposed by the progression of the COVID-19 pandemic.

Using a subjective well-being metric as an independent variable, the psychophysiological response (measured by HRV indices recorded in different postures) was compared for groups and moments. Association between subjective well-being and HRV were also investigated.

The data recorded by the authors may present valuable and novel information regarding well-being and autonomic modulation of heart rate responses to a critical condition as the strict confinement experienced by the subjects of the sample. However, some analysis lacks justification (a priori) while other methodological issues deserve some attention. Also, the discussion and conclusions sound somewhat speculative or shallow.

In general, the study may present valuable contributions to the field, but some major adjustments are proposed hoping to contribute to the production of an even better manuscript.

Abstract

Considering the suggestions made in the methods and results sections that can significantly change the content of the manuscript body, it may not be worthwhile to address the abstract section in detail now. However, here are some suggestions:

I think that the readability could be improved if the acronyms (“BSL”, “CFN” and “RCV”) were placed next to the related terms. Since it is a personal impression, please, feel free to adopt or not that adjustment.

A brief sample characterization could be appreciable in this section, also a succinct presentation of the analysis made may help the reader to orient himself in relation to the reported results.

The acronyms “WB+” and “WB-” appear in the results without properly previous presentation. Please, provide a brief explanation for what they mean.

Its not clear in the abstract (as also in the manuscript body) of which moment (BSL, CFN, or RCV) the correlations refer to. Please, clarify that aspect.

Regarding the abstract conclusions, I suggest that the authors stick only to what the results allow to conclude. To assume that 20% of the subjects had an improvement in the autonomic status goes somewhat beyond the possibilities that the results of the manuscript allow. The changes in the average values of RMSSD and HR of the two groups in the three periods in which the recordings were made can reflect a number of things, including the return to a normal status, considering an anticipatory stress effect that would already be present in some individuals of the sample, in view of the progression of the COVID-19 disease in the world (which starts in December 2019). Also, the group division, without previous justification, anchored in a metric, apparently without prior validation, raises some concerns. In addition, the arbitrary determination of a value "> 2" as a cutoff point raises another concern. The conclusion is based mainly on the results that come as consequences of these decisions that were at first not justified, and this greatly weakens the assertiveness and weight of the sentences.

At the end of the adjustments resulting from this review, I suggest that the authors rethink the abstract and, if possible, adopt a more cautious language taking into account what can actually be concluded in view of the methodological characteristics and results presented by the study.

Introduction

PARAGRAPH 1:

The first part of the first paragraph develops well. When it states: “resulted in decreased physical activity”, I suggest adopting a more measured language. Although we agree that confinement may have resulted in a reduction in physical activity, the reference cited does not supports the statement as it is made. In the next sentence, data from the national institute “Santé Publique France” are presented, I suggest including this source properly in the references and then citing it. At the end of the first paragraph the authors mention that the period of confinement may have caused changes in “cardiovascular fitness and psychological states”, I suggest complementing this sentence with other references such as: Brooks et al., 2020 (https: //pubmed.ncbi.nlm. nih.gov/32112714/), and Wang et al., 2020 (https://www.ncbi.nlm.nih.gov/pmc/articles/PMC7084952/); as also mentioning the impact on metabolism and body composition (fat mass and free fat mass) among the possible consequences of confinement, as noted in Martinez-Ferran et al., 2020 (https://pubmed.ncbi.nlm.nih.gov / 32466598 /).

PARAGRAPH 2:

The first sentence of the second paragraph could be improved if the references cited were complemented, especially with regard to the exercise's ability to mitigate the psychological effects of confinement, since Pareja-Galeano (2015 - https: //pubmed.ncbi.nlm.nih. gov / 25730691 /) addresses in his letter to the editor the interaction between types of exercises and their possible effects specifically on the symptoms of depression in response to a study that investigates the relationship between PA and symptoms of depression in adolescents.

I also suggest to complement the second sentence of the same paragraph addressing applications of HRV in contexts and populations more related to the theme of the present study as in Thayer et al., 2010 (https://pubmed.ncbi.nlm.nih.gov/19910061/), Hynynen et al., 2011 (https://pubmed.ncbi.nlm.nih.gov/20972879/), Laborde et al., 2017 (https://pubmed.ncbi.nlm.nih.gov/28265249/), and Kim et al., 2018 (https://pubmed.ncbi.nlm.nih.gov/29486547/) in addition to athletes, as in the references already adopted.

Still on this paragraph, I suggest to rethink some sentences referring to the LF HRV index, in the light of the references suggested above, as well as those that follow: Goldberger 1999 (https://pubmed.ncbi.nlm.nih.gov/10199852/), Medeiros et al., 2018 (https://pubmed.ncbi.nlm.nih.gov/29619594/), and Hayano & Yuda 2019 (https://pubmed.ncbi.nlm.nih.gov/30867063/). Focusing the paragraph on the most reliable indexes and that present better physiological description such as RR, RMSSD and HF may provide a safer way to discuss the results. In addition, the part that refer to the use of HRV for performance related ends do not seems to have a direct connection with the theme of the study, please, better elucidate this connection or suppress that part.

Finally, the paragraph ends with a statement that cannot be supported by the cited reference, which presents only associations between HRV (adopted a priori as an indicator of stress) and measures of PA, Cardiorespiratory Fitness, and body composition. It is a cross-sectional study with a sample of men only, which is not a validation study and that does not involve investigating the reliability of HRV. This sentence must be rewritten, and other references can be adopted for this purpose (Thayer et al., 2010 https://pubmed.ncbi.nlm.nih.gov/19910061/; Kim et al., 2018 https: // pubmed. ncbi.nlm.nih.gov/29486547/).

PARAGRAPH 3:

Please provide an adequate justification for the present study. Adopt also a more cautious language regarding the objectives since the methods and other characteristics of the study do not allow some of the author's purpose objectives.

I’m afraid that there is no evidence for expecting that part of the sample of the investigated population would “appreciate the confinement period”, it is difficult to conceive that this hypothesis could be postulated a priori. Please, provide a rationale for that, otherwise adjust that part.

Method

Please verify the first line of the study design subsection, it’s possible that the word “were” has been suppressed between the words “rules” and “applied”, verify if that sentence is correctly worded.

Experimental design:

I’m afraid that some information regarding the study design are missing in that subsection, while some are placed in other subsections and may better apply to the current. I would suggest the authors to offer a broader and clearer description of the study. Please, in addition to the information regarding the HRV recording periods state the other metrics adopted (VAS) and a concise declaration of the analyses made with those metrics (e.g. that the moments were compared and associations were tested) to better guide readers through the manuscript section.

Participants:

Have the authors a good reason to not include subjects with more than 60 years of age? It would be interesting provide a rationale to that age range inclusion criteria adoption. Also, regarding the inclusion criteria, please explain how the data was collected, clarify which criteria were just declared by the participants and which were really measured. Please, justify or clarify why at least forty orthostatic tests were adopted as inclusion criteria during the period of the study, if a total of just thirty where necessary, as you used 10 tests in each period for comparisons.

Heart rate variability:

Since the inter-beat intervals (IBI) were self-recorded, please, state explicitly what instructions were given to participants and if any prior training has been offered for the subjects to properly perform the IBI recordings. Please, include the validation studies regarding the IBI recording devices and for the smartphone app adopted for the IBI recording. Otherwise, mention that in the study limitations.

Regarding the artifact corrections, please, declare if any error percentage were adopted as exclusion criteria for IBI recordings and which error percentage it was. Otherwise, provide a rationale for it, since some error percentages may profoundly affect the reliability of the IBI recording and resulting HRV values (Peltola, 2012 https://pubmed.ncbi.nlm.nih.gov/22654764/). Please, also declare the recording time window adopted in each position for the HRV indices calculation and, if some stabilization period was adopted, include that information too.

Finally, please, state how did you handled with the following aspects that could acutely interfere in the HRV recordings: alcohol consumption, quality and duration of sleep, effects of vigorous exercises and other known factors. If those issues were not properly addressed, please, declare in the study limitations.

Visual analogic scale:

Please, provide the validation study that support the adopted well-being metric. Otherwise, mention that in the study limitations. Also, clarify what means the graduation values 0 to 10 that the VAS referred to, and declare if the subjects received any prior training to use that mobile application.

Statistical Section

Importantly: Regarding the group effect division adopted here (“WB+” and “WB-”), no rationale or justification was presented a priori for that division. So, the authors must explain in the manuscript why that division was adopted, which criteria were adopted for the group division and justify them, preferably based on the subject literature. I would like to add that some other group divisions, more evidently justifiable by literature, as by age clusters or by sex could be made instead of the adopted by the authors, since Brooks et al., 2020 (cited in the present manuscript references) highlighted that gender and age may be predictors of negative psychological impacts of quarantine and Hnatkova et al., 2019 (https://pubmed.ncbi.nlm.nih.gov/31611089/) showed that heart rate responses to postural changes may be different for sex.

Please, present the statistical power achieved with that sample for the analysis used.

The authors must declare if unitary or averaged values of the individuals HRV recordings were adopted in the different moments (BSL, CFN and RCV) for the comparisons made in the two-way ANOVA analysis. Also, regarding both metrics (HRV and VAS), clarify which values (unitary or averages) and of which moments (BSL, CFN and RCV) were adopted for the calculation of Pearson’s correlation coefficient. Please, provide information in a more detailed way in this topic, if possible.

Results

If some data is placed in tables it could improve that section.

Please, present the HR and HRV indices means in each period (BSL, CFN and RCV) per condition (SU and ST). Please, present the correct units in the figures 1 and 2. Also, in figure legends state of which recording positions they refer to.

Figure 3 is missing. Regarding the correlations, please clarify the points above mentioned in methods section. If correlations were traced in each one of the moments (BSL, CFN and RCV), please, present all Pearson’s correlation coefficients and respective confidence intervals, if possible. I would also suggest a regression analysis combining FC and HRV indices, since there are some interactions between those variables that can highlight or suppress the observable psychophysiological effects of interest (De Geus et al., 2018 https://pubmed.ncbi.nlm.nih.gov/30357862/)

Discussion

Most parts of discussion refer to the unjustified group division made by the authors and to a survey relating to behavioral aspects during lockdown as Physical Activity (PA) and exercise, that were not measured or controlled in the present investigation, while the interesting association between VAS and HRV indices was barely discussed. I would like to suggest to the authors a deeper and less speculative discussion, preferably based on the literature pointed out in the introduction and along of the reviewer suggestions.

Limitations

In addition to the limitations pointed out in other sessions, it would be prudent to comment that factors that may be associated with HRV or modulate their interactions with different stressors have not been controlled, such as: Body composition (Esco et al., 2011 https: //pubmed.ncbi.nlm .nih.gov / 21691230 /; Yi et al., 2013 https://pubmed.ncbi.nlm.nih.gov/23225799/), Cardiorespiratory Fitness (Buchheit & Gindre, 2006 https: //pubmed.ncbi.nlm. nih.gov/16501030/; Kiviniemi et al., 2017 https://pubmed.ncbi.nlm.nih.gov/29135784/) and others.

Conclusion

Some parts of the conclusion sound speculative or walk beyond the possibilities of the study results. Please, considering the reviewer suggestions in the sections above, rethink that section too.

References

Please adjust reference number two accordingly.

6. PLOS authors have the option to publish the peer review history of their article (what does this mean?). If published, this will include your full peer review and any attached files.

Reviewer #1: No

Reviewer #2: No

---

## [Author Response · Author response to Decision Letter 0]

29 Sep 2020

Reviewer #1: In this paper, the authors investigated changes in heart rate variability before, during and after the lockdown which took place for about two months in France.

We thank you for reviewing our manuscript and for providing comments and suggestions that have helped us to improve it. We have considered your remarks and made amendments when necessary in the revised manuscript. We appreciate your further perusal of the revised manuscript.

We have provided our responses to your comments that are in bold. Amended sentences are in italic with the additional wordings in red.

Data was collected according to standard procedure (morning test), and all typical metrics are ported, in the time (rMSSD) and frequency (LF HF) domains, for a decent sample of 95 individuals

Most importantly, subjective data was collected together with objective physiological data, which I find paramount in this case considering that despite what much of the literature reports, physiology needs to be contextualised with other variables, as exemplified by the fact that HRV could go either way (assuming a similar response for all individuals would be naive), and indeed this change does relate with the objective measurement of physiological stress derived via HRV analysis

The sample gets a bit small when stratified (e.g. only 5 women in the WB+ group), however as the changes are analysed relatively to their own historical data (baseline), I would consider this issue less problematic than in the case of comparing groups

Thank your for this comment, indeed we designed the analysis so that each individual is his own baseline which minimizes the potential bias when the groups have unbalanced number.

My speculation for the lack of change in the standing position might be linked to the protocol, often the orthostatic measurement creates more issues than it tries to resolve, with individuals getting anxious or too worked up by the quick change of posture, which might result in the test itself just capturing that transitory readjustment, which has little to do with baseline physiological stress - which is what the test aims at capturing. It could be helpful in this case simply to do two separate tests, maybe separated by a few minutes, but this procedure is of course less practical. Unfortunately, in the paper it is not clear if the protocol required a short period between the two phases (e.g. less than 30 seconds) or not, and therefore if this speculation makes sense. Please revise the paper including more information about the exact protocol, and also discuss this aspect if applicable at that point.

Thank you for this comment. The transition between the supine and standing phases was a few seconds (typically between 3 and 8 seconds). We agree that the standing phase induces a lot more issues than the supine one, but in our view (and before the pandemic crisis broke out) the supine phase is designed to assess the baseline physiological stress while the standing phase is designed to check for lack of activation of the sympathetic system or lack of withdrawal of the parasympathetic system. It is not designed to assess baseline physiological stress. For clarity, we added the following sentences in the Methods section in the manuscript:

The participants were walked through the test by the dedicated mobile application, they started the recording from the application, in the supine position. After five minutes the smartphone beeped, at which point the participants immediately stood and pressed a button on the application to start the recording of the standing phase. After five minutes in the standing phase the smartphone beeped again to inform that the test was over. The transition phase between the supine and the standing positions last 3 to 8 s on average. The HRV analysis does not take into account the first minute of recording in each position. 

Interesting to see also how the change in ANS activity was rather stable during CFN

I understand that this is not the focus of the paper, but it would be relevant for practitioners and users of these technologies if the authors could speculate on what is behind the differences, why a smaller group felt better (which was reflected in their ANS) and why a larger group had more issues, any how this second group could potentially make changes in case another lockdown happens, to favore a more positive response

Thank you for this suggestion. The factors influencing a positive or negative response to lockdown are numerous, complex, and often interact between each other. In addition, most factors are out of hands of the practitioners. We can only speculate that encouraging physical activity (but still respecting the lockdown rules) even simply walking and reassuring the population with regards to the numerous stressors (financial loss, fear of infection, lo of social activities etc.) may be the most effective ways to counteract the deleterious effects of lockdown.

Given that the data was collected as part of other studies, I wonder if any other potential correlates were collected (physical activity apparently not), but anything else that could shed some light on factors behind these differences would be relevant)

Thank you for this excellent suggestion, but unfortunately, we do not have anything substantial that could shed light on the factors explaining the differences between the groups. The strict lockdown made impossible any visit to the laboratory so that no complementary measures were recorded.

In this context I find the discussion section quite limited by the focus on physical activity. In my experience, large changes in heart rate and rMSSD baseline are hardly associated to changes in physical activity alone, especially over rather short time periods and in healthy individuals, while psychological stressors might have a much larger impact. Given that the lockdown is clearly a huge psychological stressor, I would expect more discussion points around this aspect.

We added the following paragraph in the discussion section:

Beyond this speculative influence of physical activity, there are many stressors during lockdown [2]: subjective (i.e., poor information on the duration of the lockdown, the fears of infection, frustration and boredom, sense of isolation, anger , guilt, helplessness, loneliness, nervousness, sadness) or objective (i.e., inadequate supplies, financial concerns, altered usual day-to-day activities etc.) sources of stress have an impact on psychological outcomes and indirectly on HRV.

Note also how your own data seems to highlight how the psychological aspect is key (how are you feeling is the question, which does not seem to be tuned to physical activity), more than the physical activity aspect. Of course lack of physical activity as per se a strong psychological effect especially in previously active people, but then the conversation is not about de-training, it's again about psychological wellbeing when unable to perform physical activity. Hopefully this makes sense to the authors and the discussion can be extended to better cover these aspects, which of course are all speculative since no data has been collected on other covariates 

We entirely agree with these suggestions, we have added the following in the discussion section

Through questionnaire in the application and for each test, the participants declared their potential lack of sleep, alcohol consumption (as limited as possible), digestive problems and other potential confounding factors that may have modified the HRV parameters reported in this study. Those factors are known as key in HRV and well-being alterations and may partly explain the cardiovascular deconditioning observed. However, in the present study, we did not find any factor that may have explained the differences between the two groups.

 

Reviewer #2: The manuscript presents recordings of heart rate variability (HRV) and subjective well-being in a sample of French healthy adults before, during and after a period of strict confinement, imposed by the progression of the COVID-19 pandemic.

Using a subjective well-being metric as an independent variable, the psychophysiological response (measured by HRV indices recorded in different postures) was compared for groups and moments. Association between subjective well-being and HRV were also investigated.

The data recorded by the authors may present valuable and novel information regarding well-being and autonomic modulation of heart rate responses to a critical condition as the strict confinement experienced by the subjects of the sample. However, some analysis lacks justification (a priori) while other methodological issues deserve some attention. Also, the discussion and conclusions sound somewhat speculative or shallow.

In general, the study may present valuable contributions to the field, but some major adjustments are proposed hoping to contribute to the production of an even better manuscript.

We thank you for reviewing our manuscript and for providing comments and suggestions that have helped us to improve it. We have considered your remarks and made amendments when necessary in the revised manuscript. We appreciate your further perusal of the revised manuscript.

We have provided our responses to your comments that are in bold. Amended sentences are in italic with the additional wordings in red.

Abstract

Considering the suggestions made in the methods and results sections that can significantly change the content of the manuscript body, it may not be worthwhile to address the abstract section in detail now. However, here are some suggestions:

I think that the readability could be improved if the acronyms (“BSL”, “CFN” and “RCV”) were placed next to the related terms. Since it is a personal impression, please, feel free to adopt or not that adjustment.

Thank you for this suggestion the abstract now reads:

…on a regular basis before (BSL), during (CFN) and after (RCV) the lockdown.

A brief sample characterization could be appreciable in this section, also a succinct presentation of the analysis made may help the reader to orient himself in relation to the reported results.

95 participants were included in this study (27 women, 68 men, 37 ± 11 years, 176 ± 8 cm, 71 ± 12 kg),

The acronyms “WB+” and “WB-” appear in the results without properly previous presentation. Please, provide a brief explanation for what they mean.

the methods section of the abstract now reads

The participants were split in two groups, those who reported an improved well-being (WB+, increase >2 in VAS score) and those who did not (WB-) during CFN.

Its not clear in the abstract (as also in the manuscript body) of which moment (BSL, CFN, or RCV) the correlations refer to. Please, clarify that aspect.

You are right. We modified the abstract as follows: 

When pooling data of the three phases, there was a moderate significant correlation

New paragraph was also added in the manuscript body:

Correlations

Correlations were computed for LF, HF, HR and RMSSD against VAS, in the supine and the standing positions, regardless of the time points (BSL, CFN and RCV), which means that all data points were pooled when computing the correlations. Correlations were made using MATLAB® (R2019a, MathWorks, Natick, MA, USA)

Regarding the abstract conclusions, I suggest that the authors stick only to what the results allow to conclude. To assume that 20% of the subjects had an improvement in the autonomic status goes somewhat beyond the possibilities that the results of the manuscript allow. The changes in the average values of RMSSD and HR of the two groups in the three periods in which the recordings were made can reflect a number of things, including the return to a normal status, considering an anticipatory stress effect that would already be present in some individuals of the sample, in view of the progression of the COVID-19 disease in the world (which starts in December 2019). 

We entirely agree with this suggestion and we have turned down the tone of the discussion and conclusion accordingly.

Also, the group division, without previous justification, anchored in a metric, apparently without prior validation, raises some concerns. In addition, the arbitrary determination of a value "> 2" as a cutoff point raises another concern. The conclusion is based mainly on the results that come as consequences of these decisions that were at first not justified, and this greatly weakens the assertiveness and weight of the sentences.

You are right. We toned down our conclusion.

Our results suggest that the strict COVID-19 lockdown likely had opposite effects on French population as 20% of participants improved parasympathetic activation (RMSSD, HF) and rated positively this period

At the end of the adjustments resulting from this review, I suggest that the authors rethink the abstract and, if possible, adopt a more cautious language taking into account what can actually be concluded in view of the methodological characteristics and results presented by the study.

As recommended, we modified the conclusion of the abstract, as follows:

The observed recordings may reflect a large variety of responses (anxiety, anticipatory stress, change on physical activity…) beyond the scope of the present study. However, these results confirmed the usefulness of HRV as a non-invasive means for monitoring well-being and health in the general population

Introduction

PARAGRAPH 1:

The first part of the first paragraph develops well. When it states: “resulted in decreased physical activity”, I suggest adopting a more measured language. Although we agree that confinement may have resulted in a reduction in physical activity, the reference cited does not supports the statement as it is made. In the next sentence, data from the national institute “Santé Publique France” are presented, I suggest including this source properly in the references and then citing it. At the end of the first paragraph the authors mention that the period of confinement may have caused changes in “cardiovascular fitness and psychological states”, I suggest complementing this sentence with other references such as: Brooks et al., 2020 (https: //pubmed.ncbi.nlm. nih.gov/32112714/), and Wang et al., 2020 (https://www.ncbi.nlm.nih.gov/pmc/articles/PMC7084952/); as also mentioning the impact on metabolism and body composition (fat mass and free fat mass) among the possible consequences of confinement, as noted in Martinez-Ferran et al., 2020 (https://pubmed.ncbi.nlm.nih.gov / 32466598 /).

Thank you for those very relevant suggestions, first paragraph now reads

, may have resulted in decreased physical activity.

[…]

 . Beyond these self-declared data, the impact of the lockdown period remains unclear, strong psychological impacts have been reported in a review [2] and specifically to the COVID-19-related lockdown [3], as well as metabolic and body composition outcomes [4], but one may hypothesize that it may have led to changes in cardiovascular fitness and psychological states. 

PARAGRAPH 2:

The first sentence of the second paragraph could be improved if the references cited were complemented, especially with regard to the exercise's ability to mitigate the psychological effects of confinement, since Pareja-Galeano (2015 - https: //pubmed.ncbi.nlm.nih. gov / 25730691 /) addresses in his letter to the editor the interaction between types of exercises and their possible effects specifically on the symptoms of depression in response to a study that investigates the relationship between PA and symptoms of depression in adolescents.

The beginning of paragraph 2 now reads:

Exercise plays a fundamental role in cardiovascular health [5] but also in mitigating the psychological impacts of lockdown [6]. Aerobic exercises such as walking or endurance running appears to be more beneficial than sprint running or force/power exercises [6,7].

I also suggest to complement the second sentence of the same paragraph addressing applications of HRV in contexts and populations more related to the theme of the present study as in Thayer et al., 2010 (https://pubmed.ncbi.nlm.nih.gov/19910061/), Hynynen et al., 2011 (https://pubmed.ncbi.nlm.nih.gov/20972879/), Laborde et al., 2017 (https://pubmed.ncbi.nlm.nih.gov/28265249/), and Kim et al., 2018 (https://pubmed.ncbi.nlm.nih.gov/29486547/) in addition to athletes, as in the references already adopted.

The sentence now reads:

Heart rate variability (HRV) is a commonly used method for cardiovascular follow-up in athletes [8], healthy people [9] cardiovascular patients [10], and is related to self-regulations at the cognitive, emotional, social and health levels [11,12].

Still on this paragraph, I suggest to rethink some sentences referring to the LF HRV index, in the light of the references suggested above, as well as those that follow: Goldberger 1999 (https://pubmed.ncbi.nlm.nih.gov/10199852/), Medeiros et al., 2018 (https://pubmed.ncbi.nlm.nih.gov/29619594/), and Hayano & Yuda 2019 (https://pubmed.ncbi.nlm.nih.gov/30867063/). Focusing the paragraph on the most reliable indexes and that present better physiological description such as RR, RMSSD and HF may provide a safer way to discuss the results. 

We agree and have modified the paragraph as follows:

Specifically, HRV analysis allows to evaluate the modulation of the sympathetic and parasympathetic branches of the autonomic nervous system. The low-frequency (LF) band reflects a mix of sympathetic and parasympathetic modulations on the heart [13] and has been proposed as an index of the sympathovagal balance [14]. The migration of the respiratory sinus arrhythmia in the low-frequency band is a subject of debate [15], which leaves the physiological implications of this frequency band unclear [16,17]. Respiratory sinus arrhythmia is the main phenomenon provoking changes in the HF band and RMSSD which therefore mainly reflects parasympathetic influences on the heart [18].

In addition, the part that refer to the use of HRV for performance related ends do not seems to have a direct connection with the theme of the study, please, better elucidate this connection or suppress that part.

The sentences and references linked to performance have been removed

Finally, the paragraph ends with a statement that cannot be supported by the cited reference, which presents only associations between HRV (adopted a priori as an indicator of stress) and measures of PA, Cardiorespiratory Fitness, and body composition. It is a cross-sectional study with a sample of men only, which is not a validation study and that does not involve investigating the reliability of HRV. This sentence must be rewritten, and other references can be adopted for this purpose (Thayer et al., 2010 https://pubmed.ncbi.nlm.nih.gov/19910061/; Kim et al., 2018 https: // pubmed. ncbi.nlm.nih.gov/29486547/).

Finally, HRV-based methods may be used to assess autonomic imbalances, diseases [10], stress and well-being [12].

PARAGRAPH 3:

Please provide an adequate justification for the present study. Adopt also a more cautious language regarding the objectives since the methods and other characteristics of the study do not allow some of the author's purpose objectives. I’m afraid that there is no evidence for expecting that part of the sample of the investigated population would “appreciate the confinement period”, it is difficult to conceive that this hypothesis could be postulated a priori. Please, provide a rationale for that, otherwise adjust that part.

We followed your recommendations modified the hypotheses.

We hypothesized that alteration in parasympathetic-related markers (i.e., decreases in RMSSD and HF, increase resting HR) would reflect a negative feeling over the period, whilst a minority of the population would report an improved subjective feeling with consequences on HRV, potentially in line with the report by Santé Publique France [1].

Method

Please verify the first line of the study design subsection, it’s possible that the word “were” has been suppressed between the words “rules” and “applied”, verify if that sentence is correctly worded.

Strict lockdown (further denoted CFN) rules were applied in France over 8 weeks, from 17 March to 11 May 2020.

Experimental design:

I’m afraid that some information regarding the study design are missing in that subsection, while some are placed in other subsections and may better apply to the current. I would suggest the authors to offer a broader and clearer description of the study. Please, in addition to the information regarding the HRV recording periods state the other metrics adopted (VAS) and a concise declaration of the analyses made with those metrics (e.g. that the moments were compared and associations were tested) to better guide readers through the manuscript section.

For clarity, we added detailed information in the ‘methods’ section

Nevertheless, data collection was performed for the purpose of this study, using recording devices distributed to the participants before the lockdown. The equipment consisted of a RR-interval-measuring chest strap and a dedicated mobile application that recorded both RR intervals from the strap and a declared visual analogic scale (VAS). 

The participants were walked through the test by the dedicated mobile application, they started the recording from the application, in the supine position. After five minutes the smartphone beeped, at which point the participants immediately stood and pressed a button on the application to start the recording of the standing phase. After five minutes in the standing phase the smartphone beeped again to inform that the test was over. At that point, the participants were asked to fill-up the VAS by sliding the cursor according to their sensation of the moment. The transition phase between the supine and the standing positions last 3 to 8 s on average. The HRV analysis does not take into account the first minute of recording in each position. 

Participants:

Have the authors a good reason to not include subjects with more than 60 years of age? It would be interesting provide a rationale to that age range inclusion criteria adoption. 

As stated in the manuscript, the lockdown was unexpected, the participants were equipped with the devices for other purposes than this study in the first place. The initially planned studies involved physical exercises with a maximum age for inclusion criteria of 60 years-old to limit the cardiovascular risks. We agree that it is a limitation regarding the effect of COVID-lockdown.

Also, regarding the inclusion criteria, please explain how the data was collected, clarify which criteria were just declared by the participants and which were really measured. 

Nevertheless, data collection was performed for the purpose of this study, using recording devices distributed to the participants before the lockdown. The equipment consisted of a RR-interval-measuring chest strap and a dedicated mobile application that recorded both RR intervals from the strap and a declared visual analogic scale (VAS). 

Please, justify or clarify why at least forty orthostatic tests were adopted as inclusion criteria during the period of the study, if a total of just thirty where necessary, as you used 10 tests in each period for comparisons.

From January 1st to May 11th (end of lockdown) there were 20 weeks. At 2 tests per week in average makes 40 tests to have a reasonable follow-up. In addition, if all the tests were performed in one of the three periods (baseline for example) there was no point comparing the periods, hence our additional criteria for at least 10 tests in each period, again to have a reasonable follow-up.

Heart rate variability:

Since the inter-beat intervals (IBI) were self-recorded, please, state explicitly what instructions were given to participants and if any prior training has been offered for the subjects to properly perform the IBI recordings. Please, include the validation studies regarding the IBI recording devices and for the smartphone app adopted for the IBI recording. Otherwise, mention that in the study limitations.

The H7 is the most used RR-recording device, the TP5+ uses the exact same technology:

Comparison of Heart-Rate-Variability Recording With Smartphone Photoplethysmography, Polar H7 Chest Strap, and Electrocardiography. Daniel J Plews, Ben Scott, Marco Altini, Matt Wood, Andrew E Kilding, Paul B Laursen. Int J Sports Physiol Perform. 2017 Nov 1;12(10):1324-1328. doi: 10.1123/ijspp.2016-0668. DOI: 10.1123/ijspp.2016-0668

The smartphone application is just a Bluetooth receiver, using the standard BLE protocol, which does not interfere in any ways with the recording. It is just a wireless system. The results would have been exactly the same if the RR intervals had been stored on the smartphone memory and downloaded to a computer using a wire. https://web.archive.org/web/20170130085524/https://www.bluetooth.com/specifications/bluetooth-core-specification

Regarding the artifact corrections, please, declare if any error percentage were adopted as exclusion criteria for IBI recordings and which error percentage it was. Otherwise, provide a rationale for it, since some error percentages may profoundly affect the reliability of the IBI recording and resulting HRV values (Peltola, 2012 https://pubmed.ncbi.nlm.nih.gov/22654764/). Please, also declare the recording time window adopted in each position for the HRV indices calculation and, if some stabilization period was adopted, include that information too.

Our quality check goes beyond just having the number/percentage of artifacts, it uses other measures such as closeness of the artifacts, etc. In any case the quality of the signals was checked to ensure that artifacts did not drastically affect the extracted HRV parameters. Those are in-house algorithms under patenting process; therefore, we cannot disclose in detail how they work.

Out of each of the 5-minute periods, the first minute was used as a stabilization period and was excluded from the analysis whilst the last four minutes were analyzed for HRV.

Finally, please, state how did you handled with the following aspects that could acutely interfere in the HRV recordings: alcohol consumption, quality and duration of sleep, effects of vigorous exercises and other known factors. If those issues were not properly addressed, please, declare in the study limitations.

Through questionnaire in the application and for each test, participants declared their potential lack of sleep, alcohol consumption (as limited as possible), digestive problems and other potential confounding factors that may have modified the HRV parameters reported in this study. Those factors are known as key in HRV and well-being alterations and may partly explain the cardiovascular deconditioning observed. However, in the present study, we did not find any factor that may have explained the differences between the two groups.

Visual analogic scale:

Please, provide the validation study that support the adopted well-being metric. Otherwise, mention that in the study limitations. Also, clarify what means the graduation values 0 to 10 that the VAS referred to, and declare if the subjects received any prior training to use that mobile application.

VAS is a widely used tool electronically

Visual analogue scales (VAS): Measuring instruments for the documentation of symptoms and therapy monitoring in cases of allergic rhinitis in everyday health care

Position Paper of the German Society of Allergology (AeDA) and the German Society of Allergy and Clinical Immunology (DGAKI), ENT Section, in collaboration with the working group on Clinical Immunology, Allergology and Environmental Medicine of the German Society of Otorhinolaryngology, Head and Neck Surgery (DGHNOKHC). Allergo J Int. 2017; 26(1): 16–24. doi: 10.1007/s40629-016-0006-7

Becare S.A. employees controlled that the participants properly used the mobile application and chest belt and responded quickly to any question from the participants using dedicated chat bots, emails or telephone.

Statistical Section

Importantly: Regarding the group effect division adopted here (“WB+” and “WB-”), no rationale or justification was presented a priori for that division. So, the authors must explain in the manuscript why that division was adopted, which criteria were adopted for the group division and justify them, preferably based on the subject literature.

We adopted the criteria of change in VAS >2, that corresponded to 20% change in the scale. We expected with such a large change to lower the intra-individual variability. As explained in ‘methods’, the analysis is based on a large number of records. 

Another important limitation of the present study is that our sample did not allow any other group divisions, evidently justifiable by literature, as by age clusters or gender that may be predictors of negative psychological impacts of lockdown. Moreover, Hnatkova et al. [27] showed that heart rate responses to postural changes may be different for sex.

I would like to add that some other group divisions, more evidently justifiable by literature, as by age clusters or by sex could be made instead of the adopted by the authors, since Brooks et al., 2020 (cited in the present manuscript references) highlighted that gender and age may be predictors of negative psychological impacts of quarantine and Hnatkova et al., 2019 (https://pubmed.ncbi.nlm.nih.gov/31611089/) showed that heart rate responses to postural changes may be different for sex.

We would have liked to be able to conduct such studies grouping age, gender and postural changes. But that required to have a priori groups matched for those parameters which obviously we did not have as the lockdown was unexpected. As stated in the limitations section, despite we had no prior intention to study the lockdown and failed to have such evenly constituted groups, we feel like we have a dataset that still shows some interesting results which should be brought to the readers. We added the following sentences in ‘limitations’ section

Another important limitation of the present study is that our sample did not allow any other group divisions, evidently justifiable by literature, as by age clusters or gender that may be predictors of negative psychological impacts of quarantine. Moreover, Hnatkova et al. (2019) showed that heart rate responses to postural changes may be different for sex.

Please, present the statistical power achieved with that sample for the analysis used.

Statistical power was added in the result section.

The authors must declare if unitary or averaged values of the individuals HRV recordings were adopted in the different moments (BSL, CFN and RCV) for the comparisons made in the two-way ANOVA analysis. Also, regarding both metrics (HRV and VAS), clarify which values (unitary or averages) and of which moments (BSL, CFN and RCV) were adopted for the calculation of Pearson’s correlation coefficient. Please, provide information in a more detailed way in this topic, if possible.

ANOVA analyses were carried out on unitary values (i.e. no average per period or otherwise). Pearson’s correlation coefficients analyses were carried out on unitary values (i.e. no average per period or otherwise).

Results

If some data is placed in tables, it could improve that section.

Please, present the HR and HRV indices means in each period (BSL, CFN and RCV) per condition (SU and ST). Please, present the correct units in the figures 1 and 2. Also, in figure legends state of which recording positions they refer to.

Figures 1 and 2 have been updated to also show the standing position. Units of the figures are normalized with respect to baseline as stated in the statistics section.

HR and HRV units were normalized with respect to BSL.

Figures legends have been updated to indicate the position.

Figure 3 is missing. Regarding the correlations, please clarify the points above mentioned in methods section. If correlations were traced in each one of the moments (BSL, CFN and RCV), please, present all Pearson’s correlation coefficients and respective confidence intervals, if possible.

Apologies, Figure 3 never existed. Table 1 has been added with all Pearson’s coefficient values.

I would also suggest a regression analysis combining FC and HRV indices, since there are some interactions between those variables that can highlight or suppress the observable psychophysiological effects of interest (De Geus et al., 2018 https://pubmed.ncbi.nlm.nih.gov/30357862/)

Here are the Pearson’s correlation data of HRV vs. HR data

 R p CI

LF SU -0.22 <0.01 -0.35 -0.08

HF SU -0.31 <0.01 -0.49 -0.10

RMSSD SU -0.52 < 0.001 -0.61 -0.41

LF ST 0.12 0.10 -0.02 0.25

HF ST 0.03 0.70 -0.11 0.17

RMSSD ST -0.28 < 0.001 -0.40 -0.15

We believe that there is no need to show this table in the manuscript. However, if the reviewer asked for its inclusion, we would do it

Discussion

Most parts of discussion refer to the unjustified group division made by the authors and to a survey relating to behavioral aspects during lockdown as Physical Activity (PA) and exercise, that were not measured or controlled in the present investigation, while the interesting association between VAS and HRV indices was barely discussed. I would like to suggest to the authors a deeper and less speculative discussion, preferably based on the literature pointed out in the introduction and along of the reviewer suggestions.

We fully agree and acknowledge your relevant suggestions that help to improve the manuscript.

We toned down the influence of physical activity. We deleted the following sentences:

These lifestyles and behaviors included a given level of physical exercise to maintain an adequate health status [23], and guarantee an active aging by reducing the risk associated diseases in older people [24].

In the present study, 80% of the participants reported an increased HR associated to decreased RMSSD and increased LF: these changes could be the consequences of a cardiovascular detraining, with decreased parasympathetic influences on the heart. The loss of outdoor activities as walking is likely of clinical importance since lockdown accelerates vicious circles between isolation, lack of physical activity, psychological stress/anxiety and impaired immunity [27]. Meanwhile, the participants had more time to exercise, but they had to do it by themselves, based on their personal knowledge and material possibilities, with extremely limited external advises and totally absent group motivation.

We also added the following paragraph on the psychological outcomes

Beyond this speculative influence of physical activity, there are many stressors during quarantine [2] : subjective (i.e., poor information on the duration of the quarantine, the fears of infection, frustration and boredom, sense of isolation, anger , guilt, helplessness, loneliness, nervousness, sadness) or objective (i.e., inadequate supplies, financial concerns, altered usual day-to-day activities..) sources of stress have an impact on psychological outcomes and indirectly on HRV.

Limitations

In addition to the limitations pointed out in other sessions, it would be prudent to comment that factors that may be associated with HRV or modulate their interactions with different stressors have not been controlled, such as: Body composition (Esco et al., 2011 https: //pubmed.ncbi.nlm .nih.gov / 21691230 /; Yi et al., 2013 https://pubmed.ncbi.nlm.nih.gov/23225799/), Cardiorespiratory Fitness (Buchheit & Gindre, 2006 https: //pubmed.ncbi.nlm. nih.gov/16501030/; Kiviniemi et al., 2017 https://pubmed.ncbi.nlm.nih.gov/29135784/) and others.

We have modified the ‘limitations’ section as follows:

Another important limitation of the present study is that our sample did not allow any other group divisions, evidently justifiable by literature, as by age clusters or gender that may be predictors of negative psychological impacts of quarantine. Moreover, Hnatkova et al. (2019) showed that heart rate responses to postural changes may be different for sex.

Through questionnaire in the application and for each test, participants declared their potential lack of sleep, alcohol consumption (as limited as possible), digestive problems and other potential confounding factors that may have modified the HRV parameters reported in this study. Those factors are known as key in HRV and well-being alterations and may partly explain the cardiovascular deconditioning observed. However, in the present study, we did not find any factor that may have explained the differences between the two groups.

Conclusion

Some parts of the conclusion sound speculative or walk beyond the possibilities of the study results. Please, considering the reviewer suggestions in the sections above, rethink that section too.

We modified the conclusion as follows:

The lockdown in France was sudden and severe, inducing an increase in many subjective and objective stressors as well as drastically limiting the possibilities for physical activities. This likely resulted in a stressful situation that may lead to negative psychological outcomes and/or a general cardiovascular detraining (i.e., decreased RMSSD and HF and increased resting heart rate in supine position) in most of the population

References

Please adjust reference number two accordingly.

Reference adjusted (now reference #1)

 

Further to the email sent to you from the academic editor, please accept our sincere apologies but request number 4 was incomplete. It should have read;

Thank you for including your competing interests statement; "I have read the journal's policy and the authors of this manuscript have the following competing interests: Nicolas Bourdillon and Sasan Yazdani are employees of becare S.A."

We note that one or more of the authors are employed by a commercial company: becare S.A.

We apologise for any confusionn this has caused and look forward to receiving your revised manuscript,

---

## [Decision Letter · Decision Letter 1]

26 Oct 2020

PONE-D-20-22573R1

Effects of COVID-19 lockdown on heart rate variability

PLOS ONE

Dear Dr. Bourdillon,

Thank you for submitting your manuscript to PLOS ONE. After careful consideration, we feel that it has merit but does not fully meet PLOS ONE’s publication criteria as it currently stands. Therefore, we invite you to submit a revised version of the manuscript that addresses the points raised during the review process.

We look forward to receiving your revised manuscript.

Kind regards,

Daniel Boullosa

Academic Editor

PLOS ONE

Reviewers' comments:

Reviewer's Responses to Questions

**Comments to the Author**

1. If the authors have adequately addressed your comments raised in a previous round of review and you feel that this manuscript is now acceptable for publication, you may indicate that here to bypass the “Comments to the Author” section, enter your conflict of interest statement in the “Confidential to Editor” section, and submit your "Accept" recommendation.

Reviewer #1: All comments have been addressed

Reviewer #2: (No Response)

2. Is the manuscript technically sound, and do the data support the conclusions?

Reviewer #1: Yes

Reviewer #2: Partly

3. Has the statistical analysis been performed appropriately and rigorously? 

Reviewer #1: Yes

Reviewer #2: Yes

4. Have the authors made all data underlying the findings in their manuscript fully available?

Reviewer #1: Yes

Reviewer #2: No

5. Is the manuscript presented in an intelligible fashion and written in standard English?

Reviewer #1: Yes

Reviewer #2: Yes

6. Review Comments to the Author

Reviewer #1: All my concerns have been addressed in the updated version of the manuscript, I do not have any further comments

Reviewer #2: Firstly, I would like to congratulate the authors by their effort in providing clarifications and incorporating adjustments throughout the manuscript.

Most of the issues previously pointed were solved, although some of them were not fully clarified. Considering that, I tried to be more specific in some requests and reinforced the importance of them for the higher quality possible of this manuscript.

Some minor aspects not highlighted in the previous review were pointed now, but mostly as suggestions for a better readability of the manuscript.

Methods section still brings me some concern, but I hope that with authors adjustments and clarifications we will turn that section even better.

Dear author,

Find below my review of your revised manuscript, divided by section:

Abstract

The adjustments made in the abstract were well done, congratulations.

However, I added some points that still needs attention, that follows:

I suppose that adjusting the following sentence to the plural would be appropriate:

“there was a moderate significant correlation between VAS and HR, RMSSD, HF, respectively, in the supine position”

Also, I noted that is a significant correlation between standing RMSSD and VAS present in table 1. Comments about that result are missing in the abstract and throughout the manuscript. Finally, I also suggest you to suppress the term “moderate” of the above affirmation or provide, for the readers (in the manuscript body), the parameters you adopted for that qualitative interpretation of the Pearson’s correlation coefficient.

Regarding the abstract’s conclusion I would suggest you rethink the ending of the final sentence, especially considering that, although broad, your sample may not represent the general population. In my opinion, simply changing “in the general population” to “in this population” may solve this issue.

Introduction

PARAGRAPH 1:

The adjustments made in the first paragraph are well done.

PARAGRAPH 2:

Regarding the second paragraph, the quotations on the first sentence (references 6 and 7) remains not supporting the affirmations as they are. However, you may solve that issue simply, including “some of the possible” after: “…but also in mitigating”; and adding “ In those cases” before: “Aerobic exercises such as walking or endurance…”. Or another way, feel free.

The second paragraph part referring to HRV indexes was really improved.

But the sentences referring to the LF HRV index still needs a little adjustment. Since band migration of the RSA is not the only issue regarding the physiological meaning of LF, I would suggest you complement the text alerting the readers about that. Also, please complement the quotation of the sentence that refers to HF and RMSSD, Reference 18 only supports the HF index physiological meaning, but not RMSSD index meaning.

Also, please, better clarify for the readers what are you trying to say with “of one or the other” in the following sentence: “Hypotonia of one or the other is a sign of cardiovascular deconditioning”. Or you may suppress “of one or another”, as the sentence “Hypotonia is a sign of cardiovascular deconditioning” sustain the idea of the previous sentence and reads better, in my opinion.

PARAGRAPH 3:

In the third paragraph the hypothesis adjustment was all right. However, the study still lacks an explicit justification, and the second aim is a little over tuned. But I think that there is no need to jeopardize the publication by focusing on this points more related to my perception. So, I will let those observations for your reflexion.

Method

Most of the adjustments made in Methods section are all right. However, some aspects still brings me concern.

Participants:

Thank you for the explanation about the age range of your sample.

To explicitly declare some information like that, in this section of the manuscript, can improve the comprehension of methodological processes that may not be obvious for the reader. So, please, consider to include something like the sentence that you wrote in the response letter: “… The initially planned studies involved physical exercises with a maximum age for inclusion criteria of 60 years-old to limit the cardiovascular risks”; or another similar statement. That would be fine.

Still regarding inclusion criteria, a request was not fully clarified. So, I will try to be more specific: You state that “physically active, healthy, non-smokers, with no known diseases and no pregnancy or lactation for women” subjects composed the sample. If all those data were self-declared, please, make a statement on the manuscript clarifying it. Or, if, for example, physical activity or health were measured by accelerometry or (previous to the pandemic state) physical examination, respectively, you may report which data was declared and which was measured, and how it was done.

Thank you for the explanation regarding the adoption of forty tests instead of thirty. Please, consider incorporating on the manuscript body part of the explanation provided in the response letter. Explicitly declaring some methodological decisions like those could reinforce the quality of your work, for the readers.

Heart rate variability:

Thank you for the information about the HRV measurement devices.

Please, make a quoted statement on the manuscript body about the validity of the adopted HRV measurement devices. You may simply adjust part of the following explanation provided in the response letter and concisely incorporate it to the manuscript:

“The H7 is the most used RR-recording device, the TP5+ uses the exact same technology:

Comparison of Heart-Rate-Variability Recording With Smartphone

Photoplethysmography, Polar H7 Chest Strap, and Electrocardiography. Daniel J

Plews, Ben Scott, Marco Altini, Matt Wood, Andrew E Kilding, Paul B Laursen. Int J

Sports Physiol Perform. 2017 Nov 1;12(10):1324-1328. doi: 10.1123/ijspp.2016-

0668. DOI: 10.1123/ijspp.2016-0668

The smartphone application is just a Bluetooth receiver, using the standard BLE

protocol, which does not interfere in any ways with the recording. It is just a wireless

system. The results would have been exactly the same if the RR intervals had been

stored on the smartphone memory and downloaded to a computer using a wire.

https://web.archive.org/web/20170130085524/https://www.bluetooth.com/specificatio

ns/bluetooth-core-specification”

Regarding the artifact corrections, please, state in the manuscript body that your app uses filtering algorithms not available to the public that were applied in that process, since that information may impact the replicability of your data.

Visual analogue scale:

Thank you for the complement to the manuscript about the remote support provided to the subjects. I also agree that VAS is a widely used tool. However, the way that your VAS was presented in the manuscript lacks some important information. I’m afraid that “their sensation of the moment”, as you state in some point, is a poor description and may have several meanings beyond the intended in the present study.

An adequate presentation of that psychometric instrument needs a more detailed description in the methods to ensure an adequate replicability of your study. So, I would like to suggest you report in this subsection of the methods some aspects as the anchor question, and the values on the horizontal line extremes, that are missing. Please, provide and incorporate to the manuscript a more detailed description of that psychometric instrument (VAS) that you adopted, since as states the article provided by the authors in the response letter: “The most important aspect of any VAS however is the question that is combined to it and not the line. The line remains the same, but the question may change.”.

Correlations:

Thank you for the inclusion of the information in this subtopic. That subtopic should be part of the Statistics subtopic, on my opinion. Also, please, clarify if you tested LF and HF in absolute units or normalized units, or both. If both were tested, you need to include the results of both in table 1.

Statistics Section

Thank you for the explanation regarding the choice of a change in VAS>2 as a criterion adopted for the group division. Concisely reporting that information in the manuscript body may improve the presentation of your method section for the reader.

Adjustments made in the manuscript regarding the other possible group divisions are well done.

Regarding the statistical power, in addition to presenting values you must declare in the statistical section which way you took to find those values. Please, clarify for the reader how those statistical power values were calculated.

I would like to add that the requests regarding statistical power, in that and in the previous review, were made to add robustness to your method, since the amount of data collected seems to ensure a high statistical power. So, adequately presenting that information in the methods (for both tests: ANOVA and Pearson’s correlation) could highlight a valuable aspect of your manuscript.

Results

Although partially complete, the previous requests regarding results section were well done.

Please, present in the manuscript the HR and HRV mean values (and SD) of the sample, in each recording position per period (BLS, CFN, and RCV) and per group in each period.

I also noted that a significant correlation between RMSSD and VAS was present in standing position (table 1), but the manuscript only mentions and discuss the associations observed in supine position. Please, adjust that throughout the text.

Thank you for your consideration about the interactions between HR and HRV. In my opinion, just reporting the mean values of HR and HRV in each position and per period could provide sufficient information for the reader.

Please, clarify which of the reported results of LF and HF are referent to absolute units, and which are referent to normalized units, as you mention that both were calculated in the methods section. In case of using only one of those, I suggest you present in methods section only what was analyzed and reported in results section.

Discussion

Thank you for considering the previous suggestions and for incorporate them in your discussion. However, I would now suggest you some minor and more specific adjustments in your discussion since I have made just a shallow approach to that part of the manuscript in my previous review:

At the ending of the second paragraph of the discussion I suggest you to be cautious with some affirmations. With that in mind, please, adjust the following sentence: “In this increasing well-being, the fact that increased RMSSD, LF and HF (associated to decreased HR) were also observed post lockdown, is a marker of an effective and lasting improvement in cardiovascular fitness.” I suggest you change “is” to “may be” before “a marker” and to change “cardiovascular fitness” to “cardiovascular health”. Also, in the subsequent sentence, I suggest you adopt other expression instead of “performance”, it may sound out of context, since the paragraph is not talking about entrainment or performance specifically, but in part about physical activity, that is a broader concept.

In the beginning of the fourth paragraph, I suggest you adjust the sentence to the plural and suppress the term “moderate”, in case you don’t have provided any parameter for that qualitative interpretation of the Pearson’s correlation coefficient previously in the manuscript. Also, please, discuss the significant correlation between VAS and RMSSD observed in standing position.

Limitations:

This section was really improved. Congratulations.

Please, change “is” to “was” in the following sentence of the limitations: “The separation criterion between the WB+ and WB- groups is arbitrary”.

Conclusion

The adjustments made in conclusion were nice. Congratulations.

In my opinion, changing: ”general cardiovascular detraining” to “cardiovascular health” may provide to the reader a more assertive sentence. Feel free to incorporate or not that suggestion.

Regarding the last sentence of the conclusion, please rethink the term “general population”, especially considering that, although broad, your sample may not represent the general population. In my opinion, simply changing “in the general population” to “in this population” may solve this issue.

Table 1:

Please report the measurement units of HR and HRV indexes in the first column of table 1.

Figures 1 and 2:

Figure legends describe the following symbols: “* p < 0.05 for difference with

BSL. # p < 0.05 for difference with CFN.”. However, in the figures 1 and 2 the symbols are one or two asterisks “*” or “(**)”. Please, clarify what the two asterisks symbol means.

7. PLOS authors have the option to publish the peer review history of their article (what does this mean?). If published, this will include your full peer review and any attached files.

Reviewer #1: No

Reviewer #2: No

---

## [Author Response · Author response to Decision Letter 1]

28 Oct 2020

Reviewer #2: Firstly, I would like to congratulate the authors by their effort in providing clarifications and incorporating adjustments throughout the manuscript.

Most of the issues previously pointed were solved, although some of them were not fully clarified. Considering that, I tried to be more specific in some requests and reinforced the importance of them for the higher quality possible of this manuscript.

Some minor aspects not highlighted in the previous review were pointed now, but mostly as suggestions for a better readability of the manuscript.

Methods section still brings me some concern, but I hope that with authors adjustments and clarifications we will turn that section even better.

We thank you for reviewing our manuscript and for providing comments and suggestions that have helped us to improve it. We have considered your remarks and made amendments when necessary in the revised manuscript. We appreciate your further perusal of the revised manuscript.

We have provided our responses to your comments that are in bold. Amended sentences are in italic with the additional wordings in red.

Dear author,

Find below my review of your revised manuscript, divided by section:

Abstract

The adjustments made in the abstract were well done, congratulations.

However, I added some points that still needs attention, that follows:

I suppose that adjusting the following sentence to the plural would be appropriate:

“there was a moderate significant correlation between VAS and HR, RMSSD, HF, respectively, in the supine position”.

Done

Also, I noted that is a significant correlation between standing RMSSD and VAS present in table 1. Comments about that result are missing in the abstract and throughout the manuscript.

Thank you for this relevant comment. We modified the sentence as follows: 

In standing position, HRV parameters were not modified during CFN but RMSSD was correlated to VAS.

Finally, I also suggest you to suppress the term “moderate” of the above affirmation or provide, for the readers (in the manuscript body), the parameters you adopted for that qualitative interpretation of the Pearson’s correlation coefficient.

Done

Regarding the abstract’s conclusion I would suggest you rethink the ending of the final sentence, especially considering that, although broad, your sample may not represent the general population. In my opinion, simply changing “in the general population” to “in this population” may solve this issue.

Done

Introduction

PARAGRAPH 1:

The adjustments made in the first paragraph are well done.

Thank you!

PARAGRAPH 2:

Regarding the second paragraph, the quotations on the first sentence (references 6 and 7) remains not supporting the affirmations as they are. However, you may solve that issue simply, including “some of the possible” after: “…but also in mitigating”; and adding “ In those cases” before: “Aerobic exercises such as walking or endurance…”. Or another way, feel free.

We followed your relevant suggestions. The sentences are now: 

Exercise plays a fundamental role in cardiovascular health [5] but also in mitigating some of the possible psychological impacts of lockdown [6]. In those cases, aerobic exercises such as walking or endurance running appears to be more beneficial than sprint running or force/power exercises [6,7].

The second paragraph part referring to HRV indexes was really improved.

But the sentences referring to the LF HRV index still needs a little adjustment. Since band migration of the RSA is not the only issue regarding the physiological meaning of LF, I would suggest you complement the text alerting the readers about that. 

The migration of the respiratory sinus arrhythmia in the low-frequency band is a subject of debate [15], and it is not the only issue regarding the physiological meaning of LF [16], which leaves the physiological implications of this frequency band unclear [17].

Also, please complement the quotation of the sentence that refers to HF and RMSSD, Reference 18 only supports the HF index physiological meaning, but not RMSSD index meaning.

We have added the reference to the task force (1996) to help the readers to understand the physiological meaning of RMSSD.

Also, please, better clarify for the readers what are you trying to say with “of one or the other” in the following sentence: “Hypotonia of one or the other is a sign of cardiovascular deconditioning”. Or you may suppress “of one or another”, as the sentence “Hypotonia is a sign of cardiovascular deconditioning” sustain the idea of the previous sentence and reads better, in my opinion.

We modified the sentence as suggested.

PARAGRAPH 3:

In the third paragraph the hypothesis adjustment was all right. However, the study still lacks an explicit justification, and the second aim is a little over tuned. But I think that there is no need to jeopardize the publication by focusing on this points more related to my perception. So, I will let those observations for your reflexion.

You are right. We toned down the sentence as follows:

..and to investigate whether changes in HRV were related to some changes in psychological states during and after this period.

Method

Most of the adjustments made in Methods section are all right. However, some aspects still brings me concern.

Participants:

Thank you for the explanation about the age range of your sample.

To explicitly declare some information like that, in this section of the manuscript, can improve the comprehension of methodological processes that may not be obvious for the reader. So, please, consider to include something like the sentence that you wrote in the response letter: “… The initially planned studies involved physical exercises with a maximum age for inclusion criteria of 60 years-old to limit the cardiovascular risks”; or another similar statement. That would be fine.

Thank you again for this excellent suggestion. We added this sentence in the manuscript.

The initially planned study involved physical exercises with a maximum age for inclusion criteria of 60 years-old to limit the cardiovascular risks.

Still regarding inclusion criteria, a request was not fully clarified. So, I will try to be more specific: You state that “physically active, healthy, non-smokers, with no known diseases and no pregnancy or lactation for women” subjects composed the sample. If all those data were self-declared, please, make a statement on the manuscript clarifying it. Or, if, for example, physical activity or health were measured by accelerometry or (previous to the pandemic state) physical examination, respectively, you may report which data was declared and which was measured, and how it was done.

We modified the sentence as follows:

All subjects self-declared being physically active, healthy, non-smokers, with no known diseases and no pregnancy or lactation for women as well as living in France.

Thank you for the explanation regarding the adoption of forty tests instead of thirty. Please, consider incorporating on the manuscript body part of the explanation provided in the response letter. Explicitly declaring some methodological decisions like those could reinforce the quality of your work, for the readers.

You are right. It would improve the quality of this work. The following sentences were added: 

More specifically, from January 1st to May 11th (end of lockdown) there were 20 weeks. With 2 tests per week in average per participant, 40 tests were performed.

Heart rate variability:

Thank you for the information about the HRV measurement devices.

Please, make a quoted statement on the manuscript body about the validity of the adopted HRV measurement devices. You may simply adjust part of the following explanation provided in the response letter and concisely incorporate it to the manuscript:

“The H7 is the most used RR-recording device, the TP5+ uses the exact same technology:

Comparison of Heart-Rate-Variability Recording With Smartphone

Photoplethysmography, Polar H7 Chest Strap, and Electrocardiography. Daniel J

Plews, Ben Scott, Marco Altini, Matt Wood, Andrew E Kilding, Paul B Laursen. Int J

Sports Physiol Perform. 2017 Nov 1;12(10):1324-1328. doi: 10.1123/ijspp.2016-

0668. DOI: 10.1123/ijspp.2016-0668.

The smartphone application is just a Bluetooth receiver, using the standard BLE

protocol, which does not interfere in any ways with the recording. It is just a wireless

system. The results would have been exactly the same if the RR intervals had been

stored on the smartphone memory and downloaded to a computer using a wire.

https://web.archive.org/web/20170130085524/https://www.bluetooth.com/specificatio

ns/bluetooth-core-specification”

We followed your recommendation and have incorporated this information in the manuscript as follows: 

The H7 is the most used RR-recording device, the TP5+ uses the exact same technology [23]. The smartphone application is a Bluetooth receiver, using the standard BLE protocol, which does not interfere in any ways with the recording.

Regarding the artifact corrections, please, state in the manuscript body that your app uses filtering algorithms not available to the public that were applied in that process, since that information may impact the replicability of your data.

We added this information as follows:

The RR intervals from the orthostatic tests were first analyzed to remove ectopic beats from the recordings, the algorithms used are currently not available to the public.

Visual analogue scale:

Thank you for the complement to the manuscript about the remote support provided to the subjects. I also agree that VAS is a widely used tool. However, the way that your VAS was presented in the manuscript lacks some important information. I’m afraid that “their sensation of the moment”, as you state in some point, is a poor description and may have several meanings beyond the intended in the present study.

An adequate presentation of that psychometric instrument needs a more detailed description in the methods to ensure an adequate replicability of your study. So, I would like to suggest you report in this subsection of the methods some aspects as the anchor question, and the values on the horizontal line extremes, that are missing. Please, provide and incorporate to the manuscript a more detailed description of that psychometric instrument (VAS) that you adopted, since as states the article provided by the authors in the response letter: “The most important aspect of any VAS however is the question that is combined to it and not the line. The line remains the same, but the question may change.”.

We understand your concern but the question was on purpose very vague as we aimed to obtain a general feeling of the subject. We did not focus on a special component of the mood as vigor, anxiety or anger. Here we display a screenshot (in French) of the VAS to clarify this point.

Since it is a limitation of the present study, we added the following sentence in the manuscript: 

Finally, the VAS allowed to only rate a “general well-being” but not any specific mood components as vigor, anxiety, anger etc. Therefore, it remains vague and incorporates probably different psychological states among participants.

Correlations:

Thank you for the inclusion of the information in this subtopic. That subtopic should be part of the Statistics subtopic, on my opinion. Also, please, clarify if you tested LF and HF in absolute units or normalized units, or both. If both were tested, you need to include the results of both in table 1.

As suggested, we moved this part to the “statistics” chapter.

We tested LF and HF only in absolute units. Since LF and HF are labelled nLF and nHF, we don’t believe that we should clarify in the manuscript.

Statistics Section

Thank you for the explanation regarding the choice of a change in VAS>2 as a criterion adopted for the group division. Concisely reporting that information in the manuscript body may improve the presentation of your method section for the reader.

Again, your relevant comment helps us to improve the manuscript. The following sentence was added:

Regarding the group effect division adopted (WB- vs WB+), the criteria was a change in VAS >2, that corresponded to 20% change in the scale. Such a large change aimed to lower the intra-individual variability.

Adjustments made in the manuscript regarding the other possible group divisions are well done.

Regarding the statistical power, in addition to presenting values you must declare in the statistical section which way you took to find those values. Please, clarify for the reader how those statistical power values were calculated.

I would like to add that the requests regarding statistical power, in that and in the previous review, were made to add robustness to your method, since the amount of data collected seems to ensure a high statistical power. So, adequately presenting that information in the methods (for both tests: ANOVA and Pearson’s correlation) could highlight a valuable aspect of your manuscript.

The statistical powers were calculated using the F-statistic values and the degrees of freedom, according to Scheffé, H. (1959), The Analysis of Variance. New York: John-Wiley. p. 227 ; and Zar, J.H. (1999), Biostatistical Analysis (2nd ed.). NJ: Prentice-Hall, Englewood Cliffs. pp. 195-196. Those references were added in the manuscript as well as a short explanation about the computations.

Statistical power SP is reported for each ANOVA performed, the computations take into account the F-statistic values (expected and observed) and the degrees of freedom [25,26].

Results

Although partially complete, the previous requests regarding results section were well done.

Please, present in the manuscript the HR and HRV mean values (and SD) of the sample, in each recording position per period (BLS, CFN, and RCV) and per group in each period.

For not being redundant with the figures 1 and 2, where the means (and SEM) are displayed, we politely disagree to report once more the values.

I also noted that a significant correlation between RMSSD and VAS was present in standing position (table 1), but the manuscript only mentions and discuss the associations observed in supine position. Please, adjust that throughout the text.

We added this information in the abstract and in the manuscript, as follows: 

Moreover, RMSSD in standing position was correlated to VAS.

Thank you for your consideration about the interactions between HR and HRV. In my opinion, just reporting the mean values of HR and HRV in each position and per period could provide sufficient information for the reader.

Please see our reply above re. the redundancy to report these values in figures and in manuscript.

Please, clarify which of the reported results of LF and HF are referent to absolute units, and which are referent to normalized units, as you mention that both were calculated in the methods section. In case of using only one of those, I suggest you present in methods section only what was analyzed and reported in results section.

The changes in parameters were analyzed using normalized values while the correlations were computed using absolute values. Sentences have been added in the Result section

All these changes are reported in normalized units.

All correlations were computed on absolute values.

Discussion

Thank you for considering the previous suggestions and for incorporate them in your discussion. However, I would now suggest you some minor and more specific adjustments in your discussion since I have made just a shallow approach to that part of the manuscript in my previous review:

At the ending of the second paragraph of the discussion I suggest you to be cautious with some affirmations. With that in mind, please, adjust the following sentence: “In this increasing well-being, the fact that increased RMSSD, LF and HF (associated to decreased HR) were also observed post lockdown, is a marker of an effective and lasting improvement in cardiovascular fitness.” I suggest you change “is” to “may be” before “a marker” and to change “cardiovascular fitness” to “cardiovascular health”.

Thank you. Done.

Also, in the subsequent sentence, I suggest you adopt other expression instead of “performance”, it may sound out of context, since the paragraph is not talking about entrainment or performance specifically, but in part about physical activity, that is a broader concept.

You are right. We modified “performance” for “responses”. 

In the beginning of the fourth paragraph, I suggest you adjust the sentence to the plural and suppress the term “moderate”, in case you don’t have provided any parameter for that qualitative interpretation of the Pearson’s correlation coefficient previously in the manuscript.

Table 1 reports Pearson’s correlation coefficient (R), p value, and 95% confidence interval (CI) in the supine and standing positions for HR and HRV parameters against VAS.

Also, please, discuss the significant correlation between VAS and RMSSD observed in standing position.

As requested, we added the following sentence: 

Most of these correlations were found when HRV was measured in supine position, except for RMSSD in standing position. This later result emphasizes the value of RMSSD as a key parameter of HRV analysis.

Limitations:

This section was really improved. Congratulations.

Thank you! We have added one more limitation on VAS

Please, change “is” to “was” in the following sentence of the limitations: “The separation criterion between the WB+ and WB- groups is arbitrary”.

Done

Conclusion

The adjustments made in conclusion were nice. Congratulations.

In my opinion, changing: ”general cardiovascular detraining” to “cardiovascular health” may provide to the reader a more assertive sentence. Feel free to incorporate or not that suggestion.

We agree and modified as suggested.

Regarding the last sentence of the conclusion, please rethink the term “general population”, especially considering that, although broad, your sample may not represent the general population. In my opinion, simply changing “in the general population” to “in this population” may solve this issue.

Done. Thank you!

Table 1:

Please report the measurement units of HR and HRV indexes in the first column of table 1.

Done.

Figures 1 and 2:

Figure legends describe the following symbols: “* p < 0.05 for difference with

BSL. # p < 0.05 for difference with CFN.”. However, in the figures 1 and 2 the symbols are one or two asterisks “*” or “(**)”. Please, clarify what the two asterisks symbol means.

We added the missing information.

* p < 0.05, ** p < 0.01 for difference with BSL

---

## [Editor Report · Decision Letter 2]

2 Nov 2020

Effects of COVID-19 lockdown on heart rate variability

PONE-D-20-22573R2

Dear Dr. Bourdillon,

We’re pleased to inform you that your manuscript has been judged scientifically suitable for publication and will be formally accepted for publication once it meets all outstanding technical requirements.

Kind regards,

Daniel Boullosa

Academic Editor

PLOS ONE
---

## [Editor Report · Acceptance letter]

4 Nov 2020

PONE-D-20-22573R2 

Effects of COVID-19 lockdown on heart rate variability 

Dear Dr. Bourdillon:

I'm pleased to inform you that your manuscript has been deemed suitable for publication in PLOS ONE. Congratulations! Your manuscript is now with our production department. 

Kind regards, 

on behalf of

Dr. Daniel Boullosa 

Academic Editor

PLOS ONE